# IoTwins: Implementing Distributed and Hybrid Digital Twins in Industrial Manufacturing and Facility Management Settings

Paolo Bellavista [†] and Giuseppe Di Modica *,[†]

Dipartimento di Informatica, Scienza e Ingegneria, Università di Bologna, Via Risorgimento 2,
40136 Bologna, Italy; paolo.bellavista@unibo.it
* Correspondence: giuseppe.dimodica@unibo.it; Tel.: +39-0512093277
† These authors contributed equally to this work.

**Abstract:** A Digital Twin (DT) refers to a virtual representation or digital replica of a physical object, system, process, or entity. This concept involves creating a detailed, real-time digital counterpart that mimics the behavior, characteristics, and attributes of its physical counterpart. DTs have the potential to improve efficiency, reduce costs, and enhance decision-making by providing a detailed, real-time understanding of the physical systems they represent. While this technology is finding application in numerous fields, such as energy, healthcare, and transportation, it appears to be a key component of the digital transformation of industries fostered by the fourth Industrial revolution (Industry 4.0). In this paper, we present the research results achieved by IoTwins, a European research project aimed at investigating opportunities and issues of adopting DTs in the fields of industrial manufacturing and facility management. Particularly, we discuss a DT model and a reference architecture for use by the research community to implement a platform for the development and deployment of industrial DTs in the cloud continuum. Guided by the devised architectures' principles, we implemented an open platform and a development methodology to help companies build DT-based industrial applications and deploy them in the so-called Edge/Cloud continuum. To prove the research value and the usability of the implemented platform, we discuss a simple yet practical development use case.

**Keywords:** digital twins; Industry 4.0; IoT; IIoT; RAMI 4.0; orchestration; TOSCA; cloud continuum; edge; predictive maintenance



## 1. Introduction

As the Internet of Things (IoT) and Big Data gain widespread adoption, Digital Twin technology is surging in popularity. According to a recent study by Markets and Markets [1], the Digital Twins (DTs) market, valued at USD 6.9 billion in 2022, is projected to soar to a staggering USD 73.5 billion by 2027, demonstrating an impressive compound annual growth rate of 60.6% over a 5-year span. The study identifies key industries poised to heavily invest in this technology, including Automotive and Transportation, Energy and Utilities, Infrastructure, Aerospace, Healthcare, and Oil and Gas. These industries are leveraging DTs in various applications, such as product design and development, predictive maintenance, performance monitoring, supply chain management, and business optimization. While larger organizations have established a robust pathway for adopting Digital Twins, there remains significant uncertainty regarding the speed and effectiveness with which Small and Medium Enterprises (SMEs) can embrace this innovative approach. In response to this challenge, the European community has initiated several programs, such as IoTwins, Change2Twin, and DigitBrain [2–4] with the goal of facilitating a swift and cost-effective adoption of Digital Twins among small enterprises.

IoTwins is a project supported by the European Union under the H2020 program. Commencing in September 2019, IoTwins successfully concluded its planned activities by August 2022. IoTwins exploits the big data available in *Industry 4.0* (I4.0) to devise smarter

and more effective approaches to predictive maintenance and operation optimization in industrial manufacturing. Likewise, IoTwins wants to leverage big data to derive descriptive insights about the operations and processes developed in facilities such as buildings, smart power grids, data centers, etc. On the basis of such descriptive information, optimization techniques can provide efficient facility management plans, operation optimal schedules, and renovation/maintenance plans.

The primary objective of the IoTwins project is to reduce the technological barriers faced by SMEs when seeking to create intelligent digital services that enhance their industrial production capabilities. Aligned with the guiding principles of I4.0, the IoTwins project aims to support the digital transition of industrial factories by providing them with a methodology and tools that leverage the potential of the DT computing paradigm and of ICT technologies such as Cloud/Edge computing, Big Data and Machine Learning (ML), while shielding the beneficiary from the complexity of adopting such technologies separately or together. The contributions of the paper are summarized as follows:

- The definition of a hybrid and distributed DT model;
- The design of a DT reference architecture draws inspiration from the RAMI 4.0 architectural model [5];
- The implementation of an open platform that adheres to the architecture's principles;
- The definition of a set of guidelines for the agile development of industrial DTs.

The paper structure is as follows. In Section 2, we introduce the background and discuss some related work. In Section 3, we provide a definition of the hybrid and distributed DT model devised within the IoTwins project. Section 4 is devoted to the description of the RAMI-inspired IoTwins reference architecture. In Section 5, we discuss some implementation details of the IoTwins platform, while in Section 6 a practical implementation and deployment of DT-based application is discussed. Finally, Section 7 concludes the work.

## 2. Background and Related Work

Despite the research around digital, high-fidelity copies of physical objects had flourished in the 1980s, the term "Digital Twins" is reported to first be used by Michael Grieves in 2003 at the University of Michigan [6]. Since its inception, this idea has increasingly garnered attention for its ability to create a digital depiction of a physical object. Although there is no unique, globally accepted definition of the DT concept, there are some aspects of it that is agreed upon by many researchers and practitioners [7]. Initially embraced in the manufacturing sector, the adoption of Digital Twins (DT) has extended into the domains of the Internet of Things (IoT) and cyber-physical systems (CPSs) [8]. Furthermore, it has captured the attention of various technical communities and professionals spanning diverse industries. These stakeholders have identified shared aspects between their existing methodologies, concepts, and needs. As a result, the concept of digital twins has evolved and expanded, leading to a diverse interpretation influenced by the particular domain and intended purpose. The literature is full of attempts to give a formal and exhaustive definition of DT [9–13]. In this article, we will stick to the definition proposed by Ref. [14], as we believe it embodies the concept of DT that best fits our aim: "A DT is a comprehensive digital representation of an individual product. It includes the properties, conditions, and behavior(s) of the real-life object through models and data. A DT is a set of realistic models that can simulate an object's behavior in the deployed environment. The DT represents and reflects its physical twin and remains its virtual counterpart across the object's entire lifecycle".

More specifically, *Industrial Digital Twins* (IDTs) refer to virtual representations of physical industrial assets, processes, and systems. An IDT serves as a dynamic, real-time digital counterpart of a physical asset or process, enabling monitoring, analysis, and optimization within a virtual environment. For a comprehensive list of IDT definitions, readers may consult Ref. [15]. Examples of industries that benefit from IDTs include manufacturing, energy (power plants, oil and gas facilities), transportation (aircraft, trains,

vehicles), healthcare (patient monitoring and treatment optimization), and more. Here is a non-exhaustive list of potential areas of application of IDTs in industrial scenarios:

- Real-time monitoring and data collection. IDTs continuously gather data from sensors and other sources in the real world. This data is used to update the virtual representation, ensuring that it closely reflects the current state of the physical asset or process.
- Predictive analysis involves utilizing historical data and advanced analytics within IDTs to forecast future behavior and anticipate potential issues. This capability facilitates proactive maintenance, ultimately minimizing downtime.
- Remote operation and control. By way of IDTs, operators can remotely monitor and control physical assets, even in challenging or hazardous environments. This proves especially beneficial for sectors such as energy, oil and gas, and manufacturing.
- Optimization and testing. IDTs enable the testing of various scenarios and configurations in a virtual environment prior to implementing changes to the physical asset. This can result in streamlined processes and a decrease in trial-and-error efforts.
- Reduced downtime and maintenance Costs. By predicting and preventing issues before they occur, IDTs can help reduce unplanned downtime and maintenance costs.
- Lifecycle management: IDTs cover the entire lifecycle of an asset, from design and development to operation and maintenance, and even decommissioning.

The authors of Ref. [15] list the enabling technologies that turn the DT paradigm into a concrete opportunity for industries to undertake the digitization process fostered by the Industry 4.0 revolution. Primarily, progress in technologies related to *data acquisition and analysis* (such as advanced wireless networks, communication protocols, and big data analytics) facilitates the creation of accurate digital representations and seamless integration with their corresponding physical entities [16,17]. *High-fidelity modeling* involves both the precise translation of raw data from the physical asset into knowledge and the integration of information generated by the virtual model, with the aim of optimizing the physical entity [18]. In that respect, AI is frequently employed to create models that utilize established inputs and outputs acquired from the real-world system, aiding in the comprehension of how physical properties interact with each other [19]. *Simulation* stands out as a significant enabler of the DTs, mainly due to the heightened value it provides through facilitating seamless real-time communication between virtual and physical assets. Simulating DT behavior opens up substantial opportunities for the mutual optimization of the virtual and physical models in terms of operational efficiency and maintenance schedules [20].

In the aim of establishing a *comprehensive framework* for constructing DTs, various endeavors have emerged to devise distinct modeling approaches. These efforts involve creating various DT models organized into methodological tiers, referred to as layers, to enable smooth information exchange between the physical and virtual realms. In the literature, commonly encountered modeling methods include the five-layer structure, six-layer structure, three-step process, and five-dimensional modeling [21]. In Ref. [22], the authors developed a DT reference model and architecture, and applied them in an industrial case. Drawing inspiration from the RAMI 4.0 reference model [5], they introduced a layered model consisting of three coordinated dimensions that encompass essential aspects of DTs, namely architecture, value life cycle, and integration. The literature also delves into efforts to devise a methodology for DT development. Ref. [23] proposed a methodology design using model-driven engineering (MDE) that strives toward being both flexible and generic. In this approach, a DT is initially conceptualized as a composition of fundamental components offering core functionalities (such as identification, storage, communication, security, etc.). Subsequently, an aggregated DT is characterized as a hierarchical composition of other DTs. A generic reference architecture based on these principles and a practical implementation methodology are put forward utilizing AutomationML [24].

In this paper, we present a DT model, a reference architecture, and a development methodology to help companies, both big and SME, leverage the potential of the DT paradigm in the aim of boosting their digitization process. The proposed DT model em-

bodies all the state-of-the-art approaches, i.e., the data-driven, the model-driven and the hybrid ones, and accounts for the possibility of having the DT distributed over the industrial cloud continuum (OT-to-Cloud). Similarly to Ref. [22], our reference IT architecture inspires to the principles of the RAMI 4.0 architecture; besides that, we propose an implementation of a software prototype of a DT-based PaaS for concrete use by the developers. Finally, we deliver a methodology and a set of practical guidelines for the implementation and TOSCA-based deployment of complex, containerized, and distributed DTs in the computing continuum.

**3. Design of a Hybrid and Distributed Digital Twins Model for Industrial Scenarios**

The process of digital transformation, driven by the Industry 4.0 revolution, is imperative for companies to navigate the formidable challenges presented by the global market. In line with this perspective, the European H2020 project *IoTwins* (https://www.iotwins.eu/, last accessed on 9 February 2024) strives to assist European SMEs in embracing digital transformation by making cutting-edge information technologies more accessible, effectively "democratizing" their use. IoTwins primarily aims to reduce the technological barriers faced by SMEs seeking to embrace Big Data-driven intelligent services. These services are designed to enable SMEs to extract valuable insights from their daily collected data and leverage this knowledge to enhance overall business performance. In the practice, IoTwins wants to deliver an open-software platform and a toolbox that manufacturers can harness to easily develop and operate Big Data-fueled, AI-powered, and Cloud/Edge-enabled industrial applications.

Among all enabling technologies called upon by IoTwins, the Digital Twins (DTs) paradigm plays a pivotal role. As a European *Innovation Action* project (projects of this kind enforce activities of prototyping, piloting, and market replication), IoTwins utilizes multiple industrial pilots to explore the possibilities and challenges associated with the adoption of Digital Twins (DTs) and other technologies by industrial players in manufacturing and facility management sectors. IoTwins seeks to advance the adoption of this paradigm by proposing a scalable DT model, aiming for easy replication across various industrial settings and verticals. IoTwins envisions a strong involvement of its industrial partners (IPs) in the definition of the DTs model. IPs are requested to provide the test-bed facility to support the DT model validation and the domain expertise to define the business requirements for the model design and implementation. The IoTwins test-beds are broken down into three categories: (i) *manufacturing* test-beds, (ii) *facility/infrastructure management* test-beds, and (iii) test-beds for in-field verification of the *replicability*, scalability, and standardization of the proposed approach, as well as the generation of new business models. Specifically, four industrial pilots within the manufacturing sector are focused on delivering predictive maintenance services. These services leverage sensor data to predict the time to failure, subsequently generating maintenance plans that optimize overall maintenance costs. Three large-scale test-beds concerning facility management cover online monitoring and optimization in IT facilities and smart grids, as well as intervention planning and infrastructure maintenance/renovation on sport facilities based on data collected by sophisticated and heterogeneous monitoring infrastructures. The five final test-beds, on the other hand, have been carefully chosen to demonstrate the replicability of the proposed IoTwins methodology in diverse industries, the scalability of the adopted solutions, and their ability to assist SMEs in developing new business models. In Table 1, the 12 IoTwins test-beds are listed along with a synthetic description of their claimed objectives.

**Table 1.** The 12 IoTwins test-beds.

| Test-Bed | Description |
|---|---|
| **Manufacturing Test-beds** | |
| TB1: Wind Turbine Predictive Maintenance | Developing a digital twin of a wind farm by aggregating simulation and Machine Learning models of single turbines for predictive maintenance. |
| TB2: Machine Tool Spindle Predictive Behavior | Developing multiple target-oriented digital twins of machine tools for the production of automotive components. |
| TB3: Predictive Maintenance for a Crankshaft Manufacturing System | Developing a digital twin for the predictive maintenance of a crankshaft manufacturing system |
| TB4: Predictive Maintenance and Production Optimization for Closure Manufacturing | Developing a digital twin for the optimization and predictive maintenance of a closure manufacturing system |
| **Facility Management Test-beds** | |
| TB5: Sport Facility Management and Maintenance | Developing a digital twin for the management of facilities involving the flow of large crowds in the Nou Camp stadium |
| TB6: Holistic Supercomputer Facility Management | Developing a digital twin for the maintenance and optimization of large computing facilities. |
| TB7: Smart Grid Facility Management for Power Quality Monitoring | Developing a digital twin for the computation and monitoring of a smart power grid's KPIs |
| **Replicability Test-beds** | |
| TB8: Patterns for Smart Manufacturing for SMEs | Defining a general and replicable smart manufacturing methodology for SMEs based on physics-based simulation |
| TB9: Examon Replication to INFN/BSC Datacentres | Defining a methodology for reuse of data center monitoring infrastructure in new and different contexts |
| TB10: Standardization/Homogenization of Manufacturing Performance | Defining a methodology for reuse of digital twins models for closure manufacturing in a wider series of machinery and other plants |
| TB11: Replicability towards Smaller Scale Sport Facilities | Defining a methodology for replicating and scaling facility management monitoring in other sport facilities. |
| **Business-Oriented Test-beds** | |
| TB12: Innovative Business Models for IoTwins PaaS in Manufacturing | Defining a methodology to validate innovative PaaS-based business models in the machine monitoring sector |

The DT model crafted by IoTwins utilizes big data and domain expert knowledge to depict a complex system (such as an industrial plant, process, or facility) along with its application-specific performance indicators. The ambition is to accurately predict the temporal evolution and dynamics of the system. Conceptually, IoTwins envisions the development of the following DT types: *simulation-based*, employing either agent-oriented

modeling or physical modeling; *data-driven*, utilizing cutting-edge ML/DL techniques; *hybrid*, combining the strengths of both physics and data modeling. A sample hybrid DT was developed in Test-bed 3. Here, in the aim of developing a predictive model of a machine part's faults, a ML model was implemented and trained with real-time data gathered from the sensors on board on the machine part. Given the scarcity of data sensed when the component ran in a near-to-faulty mode, the manufacturing company had to resort to simulation in order to generate synthetic data related to the malfunctioning of the part. Such data were then used along with real data to train a robust ML model for the prediction purpose. Furthermore, in order to cater for the computing needs of factories, as well as for the strict requirements of certain types of industrial applications, IoTwins fosters a *hierarchical distribution* and interworking of DTs that includes: (i) *IoT Twins*, which employ lightweight models of specific components, and conduct big-data processing and local control for quality management operations with a focus on low latency and high reliability; (ii) *Edge Twins*, situated at plant gateways, which offer higher-level control functionalities and orchestrate Internet of Things (IoT) sensors and actuators within a production locality. This facilitates local optimizations and promotes interoperability; (iii) *Cloud Twins*, which engage in time-consuming, typically off-line parallel simulation and deep-learning processes. They provide the edge twin with pre-elaborated predictive models, efficiently executed at production plant premises for monitoring, control, and tuning purposes.

From a methodological perspective, the whole DT model definition process envisaged two iterations: in the first iteration, requirements elicited from the industrial manufacturing and facility management IPs (say "group A") were used to design a first draft of the DT model, which was then prototyped in collaboration with the scientific partners and validated on the test-beds owned by *group A* IPs; in the second iteration, results collected from the mentioned experiments and new requirements elicited from the so-called replicability IPs (say "group B") contributed to refining the DT model, which was eventually validated on a second set of test-beds owned by *group B* IPs.

Finally, in Figure 1, we depict a graphical representation of the hybrid and distributed DT model developed within the IoTwins project. First, a DT (illustrated as a blue-filled box) is a distributed entity with the potential to encompass the entire spectrum of an industrial scenario. This spans from the remote Cloud down to the factory premises, where it can operate on Edge nodes and on field devices/PLC (labeled as "IoT"). Some samples of concrete DT-based applications have been depicted in the figure as red-filled boxes with a blue outline. Second, operating a distributed DT requires a well-designed, robust and scalable data backbone that will have to support the exchange of signals among the DT components (depicted as red hollow arrows) and shipping a big amount of data from the field to the Cloud (red-filled arrows). Finally, three types of models are allowed in this scenario: agent-based simulation, physics model simulation, and trained ML models. A DT belongs to one of these categories, or can be a hybrid implementation that mixes the simulation and the data-driven approach to achieve its goal (as is the case of test-bed 3).

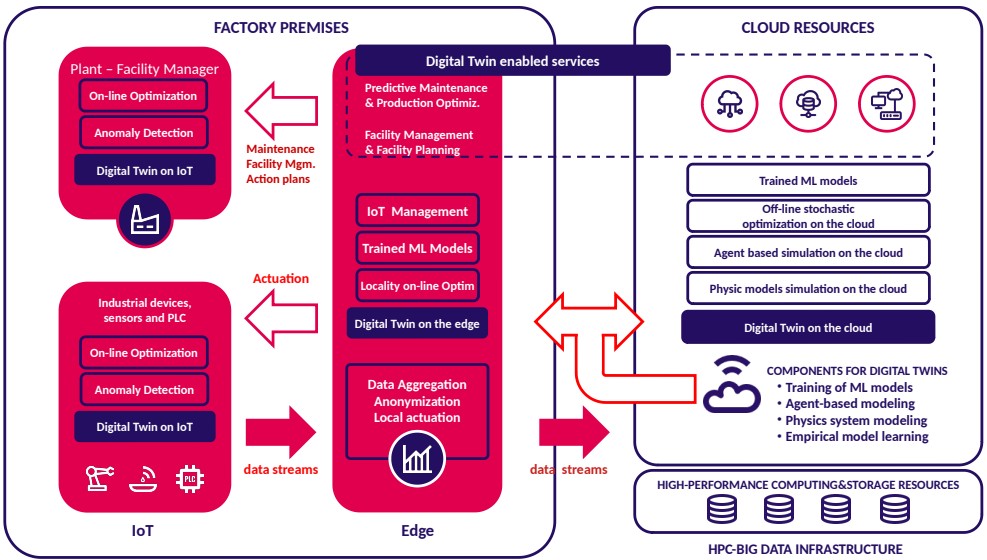

**Figure 1.** IoTwins: a distributed and hybrid Digital Twins model.

## 4. A RAMI-Inspired Reference Architecture

The *IoTwins architecture* was designed to support the DT model discussed in Section 3. According to the model specification, the supporting architecture will have to accommodate: (i) the development of digital objects modeling physical industrial entities and industrial processes; (ii) the design and deployment of pipelines for shipping data from the physical entities to the digital objects, back and forth; (iii) the deployment and operation of digital objects in a distributed and heterogeneous computing environment (i.e., the OT-to-Cloud computing continuum). First, due to the unbalance of resources in the continuum, the architecture will have to provide functions to build virtualized and uniform execution environments out of the physical computing power. Second, due to the intrinsic distributed nature of the conceived digital twins, the architecture will provide a solid mechanism and a choice of protocols to support reliable communication among the multiple digital objects that will populate the above-mentioned virtualized environment. Third, a service is needed to orchestrate the life-cycle (creation, configuration, resource provisioning, roll out, running) of all digital objects. Finally, the architecture will have to guarantee the fulfillment of strict security requirements of compute resources included in the OT perimeter (IoT and Edge ones in our scenario) and of real-time requirements imposed by mission-critical industrial applications.

The IoTwins architectural model draws inspiration from the Reference Architectural Model Industry 4.0 (RAMI 4.0) [5] developed by the German Electrical and Electronic Manufacturers' Association (ZVEI) to support Industry 4.0 initiatives. We chose to base our architectural model on the one proposed by RAMI since the latter is a well-established and world-wide recognized reference architecture to implement digital transformation processes in industrial settings, and it also already embeds many of the features that we seek to build a DT-based platform upon. RAMI 4.0 provides a unified model that ensures all the stakeholders involved in an I4.0 ecosystem share data and information in an efficient and effective way. The RAMI 4.0 model comprises three "axes" named Life Cycle value stream, Hierarchy levels, and Architecture Layers, respectively. Grounding on the IEC62890 standard [25] the Life cycle value stream axis provides a view of the product life cycle from conception to disposal. The foundation of the Hierarchy levels axis are IEC62264 [26] and IEC61512 [27] respectively, which aim to represent different functional levels of a factory. Finally, the Architecture layers axis enables the transformation of industrial assets into their interoperable Digital Twins. This axis support most of IoTwins research objectives. In the following, we report a short description of it.

The Architecture layers axis defines a framework where the physical world meets the digital one and a strong interconnection among the manufacturing operations is enabled. A layered view of the axis is depicted in the left-end of Figure 2. At the bottom, the *Assets layer* identifies and describes the real assets in the physical world. It comprises sensors, devices, machine parts, machines, machine groups, etc. The *Integration layer* describes the digital equivalents of physical assets. This layer is where the transition from the physical world to the cyber space begins. The *Communications layer* addresses mechanisms for the interoperable exchange of information between digital assets. The *Information layer* defines data services such as provisioning and integration that can be leveraged to exchange data among functions, services, and components. The *Functional layer* provides the runtime and modeling environment to build functions and services to support the business. Finally, the *Business layer* defines organizational and business-related applications, processes, and operations. Inspired by the RAMI4.0 Architecture layers axis, the *IoTwins architecture* aims to provide a reference architecture to guide the implementation of software platforms for building, operating and maintaining DT-based industrial applications. IoTwins proposes a logical, layered architecture defining the functions that the software platform will have to offer. In Figure 2, we illustrate the IoTwins architecture [2,28] and highlight the mapping between the architecture functions and the RAMI4.0 concepts that each function addresses.

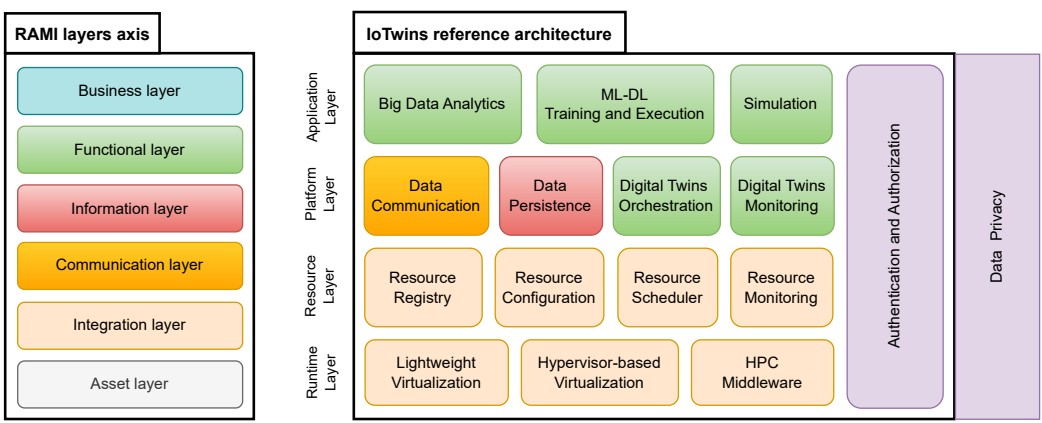

**Figure 2.** IoTwins-RAMI 4.0 mapping.

The IoTwins architecture addresses RAMI concepts ranging from the 'Integration' to the 'Functional'. The RAMI 'Asset' layer represents physical things in the shop floor (production line machines, sensors, actuators, etc.) that need to be connected to the digital world; therefore, as IoTwins is a reference architecture for software platforms, it will build on top of those assets. Similarly to the RAMI4.0 model, the IoTwins architecture adopts a layered approach, with each layer leveraging the functionalities provided by the lower layer and offering services to the upper layer. In the following, a detailed description of such layers and the RAMI4.0 concepts they address is given.

### 4.1. Runtime Layer

In order to project an industrial physical object ("thing", in the following) into the digital world, it is necessary to set up a computing environment where the digital *alter ego* of the thing can live. This layer is responsible for abstracting the available computing resources (which may range from very small IoT devices to large High Performance Computing (HPC) clusters) and providing a virtualized execution environment that can flexibly accommodate the computational demand of DTs. Technologies that may serve the mentioned purpose include lightweight virtualization, hypervisor-based virtualization, and HPC middleware.

### 4.2. Resource Layer

We refer to a "resource" as a virtual computing entity that can be activated on demand on the virtualized execution environment. A very simple form of a digital copy of a

physical thing can even be implemented by means of a simple resource (e.g., a microservice mirroring a sensor's data). This layer recommends a set of services that aim to guarantee full dependability of resources when they are operational. Among others, this layer takes care of tracking all running resources, scheduling new resources on demand, monitoring the resources status, and implementing resource resilience strategies.

### 4.3. Platform Layer

This layer offers functions to build more complex and faithful digital reproduction of any factory asset (be it an operating machine, a production line, or the supply chain process). In IoTwins, a DT is conceived as a composite digital object, consisting of a certain number of simpler digital entities capable of interacting with one another to achieve the DT business goal and of executing anywhere in the computing continuum. The Platform layer will include components that offer services to: (i) support one-to-one, one-to-many and many-to-many communication among a DT's entities and among DTs; (ii) meet the data persistence needs of the DTs; (iii) orchestrate the composition, deployment, operation, and maintenance of DTs; (iv) constantly monitor and guarantee the DT service continuity.

### 4.4. Application Layer

This layer includes application templates and toolkits to assist developers in implementing the DT business logic. In this regard, two approaches are supported for the development of DTs: a data-driven approach, which makes intensive use of ML/DL and Big Data analytics, and a model-driven approach, which relies on the use of software simulation. The support for a combined use of data-driven and model-driven techniques for the development of a hybrid DT is also provided.

### 4.5. Authentication and Authorization

This is the ingress point to access the services offered in the Platform layer and in the Application layer. Here, access procedures are put in force in order to grant a safe and controlled access to both data and services.

### 4.6. Data Security

Privacy is a strong requirement that cross-cuts all the layers. Since private and sensitive data may be handled, both raw data coming from the shop floor and those elaborated along the path must be secured. In that respect, procedures to protect data at rest (e.g., data anonymization, data encryption) as well as data in transit (e.g., secure communication channels) must be enforced.

## 5. The IoTwins Platform: A Software Prototype

We implemented a software platform that adheres to the design principles specified by the IoTwins reference architecture. As mentioned in Section 3, the platform went through two refinement iterations governed by an overarching process of (i) requirements elicitation from industrial test-beds and (ii) use case validation. In this section, we will disclose some implementation details of the platform prototype, illustrating the software environment that we instrumented to operate industrial Digital Twins in the computing continuum. Finally, we will discuss the point of view of software developers, taking advantage of the tools offered by the IoTwins platform to build and deploy a DT-based application.

### 5.1. IoTwins Platform's Implementation Details

To support the coding of the platform's prototype, many commonly available and highly mature open-source software products have been used. Considering the maturity level of such software, whose Technology Readiness Level (TRL) is in the range 8–9, and the extensive tests that all industrial partners ran to attain their goals, we can claim that the final version of the platform released at the end of the project achieved an estimated maturity level of TRL 6.

The platform is a distributed system made up of a number of software components that can be deployed in the continuum. As the reader may recall, the architectural design addresses three levels of computing environment, namely, IoT, Edge, and Cloud. The DTs model adopted in the IoTwins project was conceived to cater the need (expressed during the requirement elicitation phase) that the software components of the applications' business logic should be able to run across all computing environments (IoT, Edge, Cloud) according to the specific goal they need to accomplish, and still manage to communicate with other peer components in a seamless fashion. For example, a ML model must be run on the Cloud for training purpose, but it then needs to be migrated closer to the industrial machine (e.g., on the Edge) to make certain predictions in a responsive way. Data streamed from the shop floor can be used to feed the model both during the training session and when it makes inference. To cater for this need of facilitating the components portability and intercommunication, we designed and implemented the distributed platform in a way that one software stack could fit the three computing environments. Therefore, the three software bundles deployed on IoT, Edge, and Cloud will all look the same with only slight differences. Each has a communication layer leveraging asynchronous messaging, reusable adapters for data streaming and data storage, a virtualization layer to manage in a flexible way the underlying computing power. The software stack on the Cloud also includes the logic for orchestrating the data pipelines and workflows (Orchestrator), while the one on the IoT is equipped with plug-ins to collect data from field devices that speak different industrial protocols.

For the sake of brevity, in the following we are going to give a description of the platform components that are commonly deployed on the Cloud side and on the Edge side, as depicted in Figure 3 and Figure 4, respectively. In the final tests conducted at the end of the project, the Cloud-side platform components were deployed in the private data center of one of the technology providers that participated in the project. For what concerns the Edge components, they were deployed on a commodity PC equipped with Linux Ubuntu 20 OS.

Bottom-up, we are going to briefly discuss the software products that provide an implementation of the functions/services populating the hierarchical layers of the RAMI-inspired IoTwins architecture. In each figure, a colored legend recalls which RAMI's layer a given platform component addresses. At the bottom of Figure 3 the software tools implementing the RAMI's integration layer on the Cloud side are depicted. Virtualization software such as Openstack (https://www.openstack.org/, last accessed on 9 February 2024) Docker (https://docker.com, last accessed on 9 February 2024), and Kubernetes (https://kubernetes.io, last accessed on 9 February 2024) was employed to abstract the underlying physical computing and storage resources and offer them as a pool of virtual resources that can be managed in an easier and more uniform way. The heart of the platform is the INDIGO Orchestrator [29], which contributes to implement services belonging to the RAMI's functional layer. The INDIGO Orchestrator is a TOSCA-compliant [30] cloud orchestrator in charge of accepting application deployment requests, scheduling virtual computing resources, and enforcing provisioning workflows that serve the requests. Beside fulfilling the application deployment task, INDIGO is capable of enforcing actions that guarantee the scalability and fault-tolerance of the deployed applications. In the next section, a sample provisioning workflow enforced by the INDIGO orchestrator is described. The platform provides services to help the developer implement the data backbone of their DTs. On the left-end of the figure, the RabbitMQ message broker and a set of streaming data adapters are depicted in yellow, signifying their belonging to the RAMI's Communication layer. These components are responsible for the gathering of data from the Edge and their adaptation to the application's required format. On the right-end of the picture, some DBMS tools (MinIO (https://min.io, last accessed on 9 February 2024) as object-storage, InfluxDB (https://www.influxdata.com, last accessed on 9 February 2024) as time-series, MongoDB (https://www.mongodb.com, last accessed on 9 February 2024) as NoSQL) are depicted along with the adapters that developers can craft with the support of the

telegraf (https://www.influxdata.com/time-series-platform/telegraf/, last accessed on 9 February 2024) tool. The support for data management is then enriched with two open repositories that store Docker-based modules that can be re-used for DT development purpose. Finally, in the figure a sample of potential industrial applications is depicted: *Control logic* is a software application that implements the business logic of an industrial control loop; *Data pre-processor* is a component that filters/adapts streamed data before feeding them to the control logic; and *ML training* is a neural network that needs to be trained both on locally-stored historical data and on real-time data streamed from the Edge.

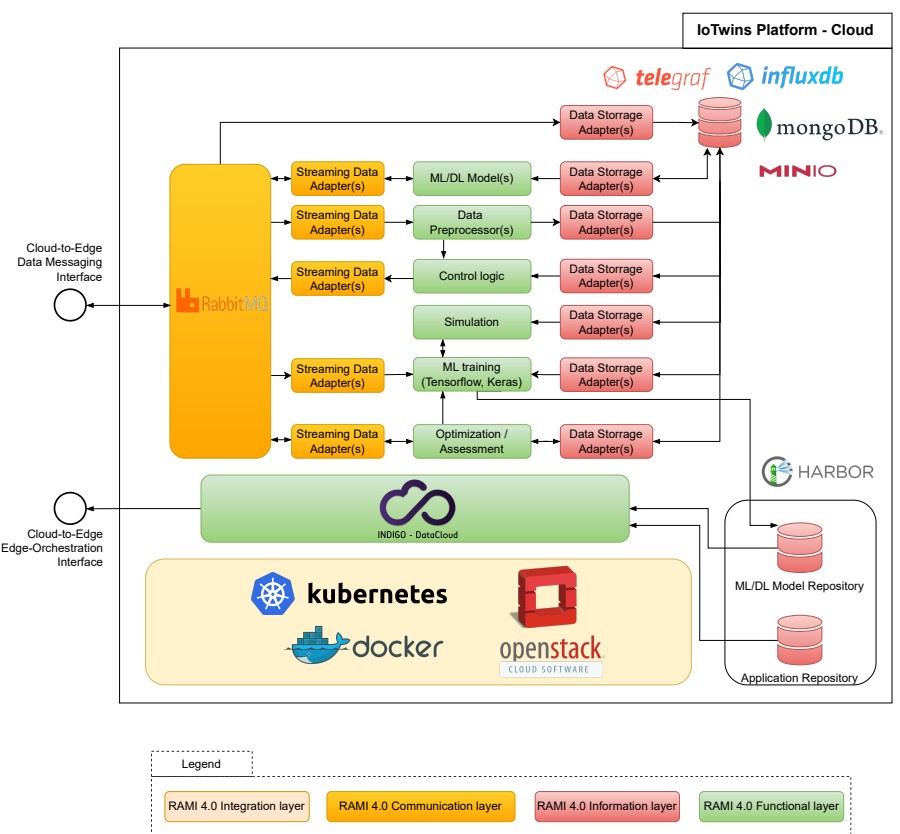

**Figure 3.** Software components of the IoTwins platform deployed in the Cloud.

In Figure 4, the software components implementing the Edge side of the IoTwins platform are depicted. In the bottom, the virtualization tools Apache Mesos (https://mesos.apache.org/, last accessed on 9 February 2024), Marathon (https://github.com/mesosphere/marathon, last accessed on 9 February 2024), and Docker implement the integration layer prescribed by the RAMI architecture. Despite that the Mesos tool is designed to virtualize and manage a cluster of computing nodes, it perfectly accomplishes the management duties of just one node. Furthermore, in future developments it will cope well with scenarios where multiple Edge nodes need to be managed. Tools such as Marathon and Chronos, in their turn, will offer the developer the opportunity of running long-running and job-like computing instances, respectively. The reader may have noticed that there is no orchestrator component deployed in the Edge. The reason is that, at design time, we decided to centralize the orchestration functionality in the Cloud, so there is just one component (the INDIGO orchestrator, indeed) responsible for orchestrating the computing resources belonging to the Cloud/Edge continuum. On the Edge side, orchestration instructions are remotely triggered by the INDIGO orchestrator to the Mesos tool via the REST interface. Similarly to the Cloud deployment, software adapters are provided to meet both data stream and data storage adaptation needs. On the left end of the picture a *Data Collector* component is displayed. It will cater for the need of collecting data from IoT devices in the field independently of the communication protocol they

use. Finally, a list of sample components is depicted in green that represent potential applications that the developer may decide to run on the Edge.

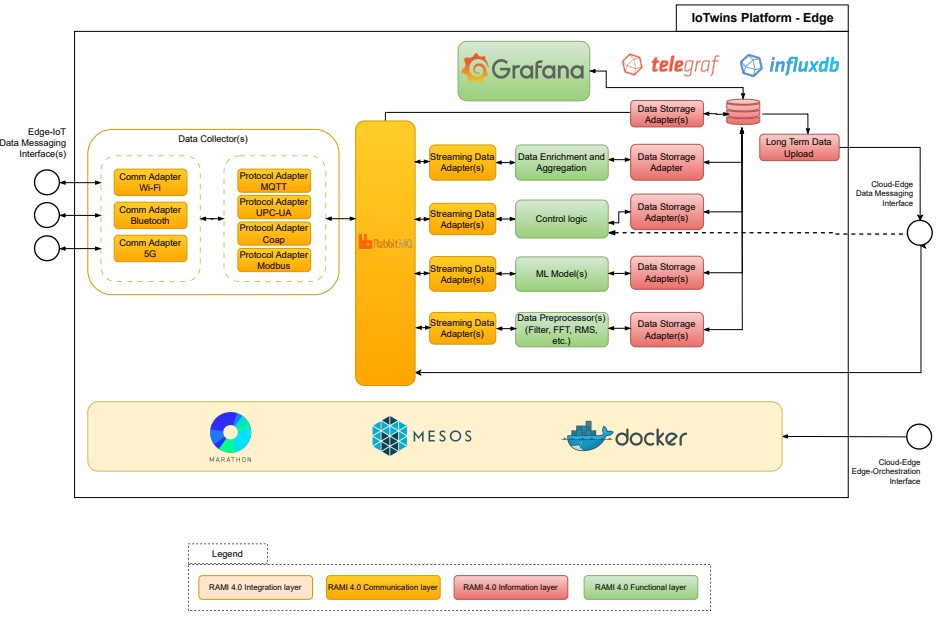

**Figure 4.** Software components of the IoTwins platform deployed in the Edge.

### 5.2. Digital Twins Implementation Guidelines

As mentioned in Section 3, the IoTwins project aimed at "democratizing" access to most advanced information technology, i.e., providing SMEs with cheap instruments to embark the digital transformation facilitated by I4.0. The IoTwins open platform is one of the most remarkable project outcomes, benefiting both the research community and industry. The platform not only allows users to easily build DT-based services to support their own businesses; it also enables service developers to implement new, re-usable software modules that will contribute to the consolidation of an ecosystem of freely accessible and composable services from which the community can draw to accelerate the development path of their DTs. To support this objective, the IoTwins initiative created an open service repository and populated it with some representative and popular services. It also created a set of rules for service developers to follow in order to design platform-compatible DTs.

In the IoTwins framework, a DT can be a very simple software module, running in either the Cloud or the Edge, or can take the form of a complex chain of interworking modules deployed and running along the continuum. Because the Docker framework is used as containerization technology in both Cloud and Edge-level runtime environments, each IoTwins DT must be developed as a (composition of) Docker container(s). This enables "coding" the DT once and deploying it anywhere along the Docker-powered Cloud/Edge continuum. Furthermore, in order to meet the INDIGO's "orchestrability" criteria, an ad-hoc TOSCA template including instructions on how to deploy the software module must be provided for each DT. We will use the term "Toskerization" to designate to the process of creating a new DT, which stems from a crasis between the phrases TOSCA and Docker.

A Toskerized DT is a software bundle that embeds a Dockerized image of the service (i.e., a service that can run in a Docker runtime environment) and a TOSCA file (indeed, the service template) that instructs the INDIGO PaaS Orchestrator on how to correctly deploy the service upon the user request. The Dockerized service can be built out of a plain Docker image publicly retrievable from any of the available Docker repositories, by optionally adding extra layers according to the specific needs. Basically, the Dockerized DT must be configured to accept a list of input parameters, that the user may want to pass for correctly configuring the service, and a list of output parameters needed to configure other

Dockerized services that might be deployed along in a service chain fashion. The described approach simplifies the service deployment operations in a consistent way. In fact, the user does not have to directly handle the software package/tool that they need, nor do they have to manipulate configuration files. The declarative approach offered by the TOSCA standard (and enforced by the INDIGO orchestrator component of the IoTwins platform) offers the user an easy way to declare what their deployment objectives are and takes care of the entire deployment process (i.e., pulling the software tools from a repository, install it in a private computing space, configure it, re-run the deployment in case of temporary failures, etc.). Service developers who want to make their DTs orchestrable by the IoTwins platform must follow a few simple principles that show how the Toskerization process works. The process includes the following steps:

1.  Creating the Docker image. The developer will have to explore public Docker repositories to search for existing dockerized images of the service that they wish to implement. If such an image is not available, they will have to build the image from scratch. When editing the Dockerfile, the developer will have to make sure that the image accepts input values for the correct configuration of the service at runtime: the easiest way to accomplish this goal is to pass input data values through C-shell environment's variables, as most ready-to-use docker images are already set to read variables from the environment where they execute; in the case that further configuration work is needed, the developer will have to create ad hoc scripts to be injected in the Docker image and run them;
2.  Uploading the Docker image to the IoTwins repository. The platform is provided with a private docker container repository that offer storing, image retrieval, and text-based search functionalities. Once uploaded on the repository, the docker image can be accessed by the orchestrator in a transparent way in order to enforce provisioning tasks;
3.  Coding the TOSCA template. The TOSCA standard offers a declarative approach to define the topology and the provisioning workflow of cloud-based and distributed application. The developer is in charge of mastering the TOSCA-compliant blueprint containing the instructions to provision the software modules implementing the DT. Instructions are declarative statements concerning, e.g., the computing capacity requested by the DT, the configuration properties of the software modules (docker components) that the DT is composed of, their mutual dependencies, etc.;
4.  Testing the DT orchestrability. In order to run functional tests on the mastered TOSCA template and on the related provisioned services, the developer can make use of two front-end tools: a command line interface (CLI) and a web-based interface. Both tools let the developer send deployment commands to the INDIGO orchestrator and monitor/debug the provisioning process in a sandbox environment.

## 6. Building and Provisioning an Industrial Digital Twin

We present an illustrative case study that delves into the definition, implementation, and deployment of a DT adhering to the IoTwins reference model and spanning the Cloud/Edge industrial continuum. The case under discussion is a tangible instance of a developmental initiative conducted within the framework of an IoTwins test-bed implementation. The collaborating partner owning the test-bed expressed the need of developing an AI-driven application for identifying irregularities within an industrial machine tool during its production process. In pursuit of this objective, the machine was outfitted with sensors designed to capture specific physical metrics (such as load, forces, vibrations, etc.). The intention was to accumulate an extensive dataset, which would subsequently be used to train a ML model proficient in detecting potential functional anomalies associated with the aforementioned machine tool. Due to the time-critical nature of the control loop (detecting potential anomalies quickly reduces the risk of tool damage), it is essential to execute the trained ML model as approximately as feasible to the data sources. While an Edge computing node satisfies this demand, it cannot ensure the computational power necessary

for ML model training. In contrast, the Cloud emerges as the more suitable computing environment, offering the requested capacity for training the ML model. Developers may encounter various technical and administrative challenges when implementing such an application. To begin, they must establish a data path across the continuum (from sensors through Edge to Cloud) to ensure a continuous flow of data to both the ML models—the one under training and the trained one. Secondly, the modules constituting the DT applications need appropriate configuration, interconnection, and deployment within a distributed computing environment. Subsequently, once the Cloud-based ML model has been successfully trained on data at rest, it must be transferred to the Edge where it will receive real-time data.

The IoTwins platform equips developers with tools and services to confront these challenges and expedite application development. Following the guidelines outlined in the previous section, developers will explore the platform's repository to identify reusable modules that align with the application's objectives. Fortunately, the repository provides a range of Docker containers for constructing the DT's data backbone. These encompass message brokers for data distribution, adaptable connectors for data format and protocol conversion, as well as databases for diverse data storage requirements. These software components can be effortlessly assembled by developers to create the desired data path. Developers are tasked with implementing the ML model and any supplementary elements pertinent to the application's business logic. These components should be containerized using the Docker framework and uploaded to the repository. When all DT components are available in Dockerized forms, developers will focus on mastering the TOSCA blueprint that governs the entire DT structure. This involves populating the blueprint with configuration parameters for the components, instructions for component interconnections, and the callback mechanism necessary for migrating the ML model component from Cloud to Edge following its training. An excerpt from the TOSCA blueprint, demonstrating the interdependencies between two components of the DT data infrastructure (specifically, the message broker and a connector), is displayed in Listing 1.

Subsequently, the developer will submit the TOSCA blueprint to the orchestrator, which is responsible for deploying the DT components based on the provided instructions. The orchestrator has the capability to deduce the correct order of deployments to be executed. In Figure 5, we present a visual representation of the DT components deployed within the cloud continuum. In this illustration, reused components are denoted in a deep blue color, while components that the developer crafted from scratch are shown in a light blue hue. The solid lines indicate the exchange of data between pairs of components, while the dashed line represents the migration path of the ML model. Some components are deployed on an Edge node within the factory premises, while others are provisioned in the Cloud. The data backbone supporting the application's logic comprises several key components:

- Message Broker (RabbitMQ): This component serves as a central hub for data exchange. It collects data generated in the field by various sources and ensures its delivery to the intended recipients.
- Data Stores:
  - InfluxDB (Time-series Database): This database is responsible for storing time-series data, which can be crucial for analyzing trends and patterns;
  - MinIO (Object Storage Database): MinIO serves as an object storage database, housing data objects and making them accessible for various purposes.
- Connectors:
  - RabbitMQ-to-InfluxDB Connector: this connector facilitates the transfer of data from the message broker to the InfluxDB, enabling data to be stored for further analysis;
  - InfluxDB-to-MinIO Connector: this connector assists in moving data from InfluxDB to MinIO, possibly for archival or other use cases.

**Listing 1.** Exert of a TOSCA blueprint for the provisioning and wiring of a RabbitMQ docker instance and a Telegraf connector instance.

```
topology_template:
  inputs:
#params for RabbitMQ, Telegraf and InfluxDB
#....omitted code.....#

node_templates:

    rabbitmqnode:
      type: tosca.nodes.indigo.Container.Application.
                  Docker.Marathon
#....omitted code.....#
      artifacts:
        image:
          file: rabbitmq:management
          type: tosca.artifacts.Deployment.Image.
                  Container.Docker
      requirements:
      - host: rabbitmqdockerruntime
    rabbitmqdockerruntime:
      type: tosca.nodes.indigo.Container.
                  Runtime.Docker
#....omitted code.....#
    telegrafnode:
      type: tosca.nodes.indigo.Container.
                  Application.Docker.Marathon
      properties:
        environment_variables:
          RABBITMQ_ENDPOINT:
            type: string
            value:
             {concat: [ "mqtt://",
                {get_attribute :
                  [rabbitmqnode,
                    load_balancer_ips, 0 ] }, ':',
                {get_attribute :
                  [rabbitmqdockerruntime,
                host, publish_ports, 1, target]}]}
#....omitted code.....#
      artifacts:
        image:
          file: iotwins-harbor.cloud.cnaf.infn.it/
            data-services/
            iotwins-connector-rabbitmq-influxdb:1.0
          type: tosca.artifacts.Deployment.
                      Image.Container.Docker
      requirements:
        - host: telegrafdockerruntime
    telegrafdockerruntime:
      type: tosca.nodes.indigo.Container.
                  Runtime.Docker

#....omitted code.....#
```

The data generated in the field is first collected by the message broker and then routed to its intended destinations through these components. On the Cloud side, the ML model component, depicted as the "Anomaly Detection (AD) Model", retrieves data from the MinIO object storage. This data is essential for the training process of the ML model, which is a critical part of anomaly detection. On the Edge end, the trained AD model component subscribes to the message broker in order to receive fresh data generated by the sources. Based on those data, it will have to detect potential anomalies and notify with the Alarm component. Finally, the Data Polisher filter and cleans in-transit data before they get to the Cloud.

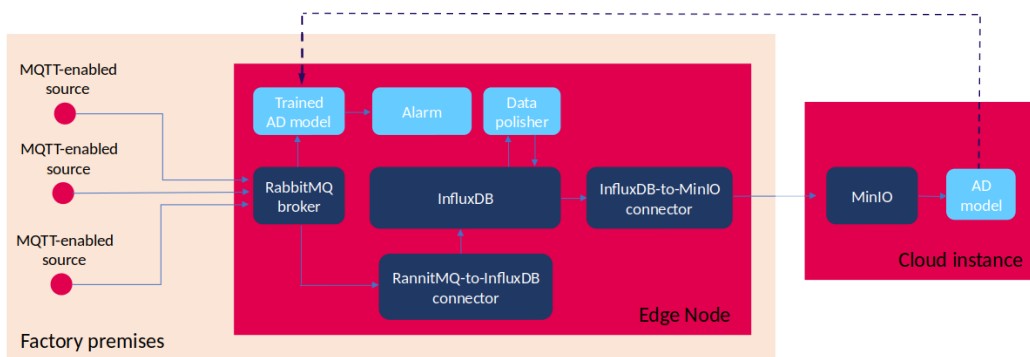

**Figure 5.** Digital Twins software components provisioned by the orchestrator.

## 7. Conclusions

This paper provides an overview of Digital Twins reference model as presented in the EU-funded IoTwins project. In particular, the paper focuses on the design and development of a software prototype of an open platform to support the agile implementation of Digital Twins-based applications in industrial settings. One of the factors that contributed to the success of the IoTwins project is the enthusiastic participation of several players coming from diverse sectors of the industry. In the first phase of the project, a doubly-iterated requirements elicitation process was triggered to assess and refine the specific needs of the involved industrial test-beds. The variegated set of functional and non-functional requirements contributed to the definition of a hierarchical and hybrid Digital Twins model; then, it guided both the design of the IoTwins architecture and the implementation of the open-source platform prototype that can be adopted in industrial settings for the development of digital twins-based application. One more remarkable objective attained by the project is the tear down of cultural and economical barriers that prevent stakeholders from fully leveraging the relevant technology adopted within the project. To this end, the project developed a set of development guidelines that will help developers to build Digital Twins by composing existing containerized software. In the final part of the project, intensive tests conducted by the industrial partners in real industrial settings proved that operators with medium-level IT skills successfully implemented and operated distributed Digital-Twins-based applications by simply leveraging the tools offered by the IoTwins platform.

Overall, this work contributes to the state of the art in the field by proposing a RAMI-inspired Digital Twins reference architecture and delivering a novel and practical "build-by-compose" approach and easy-to-use tools to the development of industrial applications in the *industrial continuum*, i.e., the environment spanning the whole chain of computing resources ranging from the shop floor to the remote cloud. The positive achievements attained in IoTwins, encompassing improved time-to-market of digital twin applications and decreased financial investment, mark a substantial advancement. We claim that these achievements will contribute to motivate SMEs to expedite the digitazation processes fostered by Industry 4.0. As future work, we are going to enhance the DevOps tools delivered by the project in order to streamline the time-to-production and operation of distributed industrial applications that leverage the Digital Twins paradigm. While doing

so, we also intend to extend the use of the platform in other industrial fields to further refine the model and the architecture devised by the project.

**Author Contributions:** Methodology, P.B.; Investigation, G.D.M.; Writing—original draft, G.D.M.; Writing—review & editing, P.B. All authors have read and agreed to the published version of the manuscript.

**Funding:** This work was partially supported by the EU H2020 IoTwins Innovation Action project (g.a. 857191).

**Data Availability Statement:** The original contributions presented in the study are included in the article. Further inquiries can be directed to the corresponding authors.

**Conflicts of Interest:** The authors declare no conflict of interest.

## Abbreviations

The following abbreviations are used in this manuscript:

| | |
|---|---|
| DTs | Digital Twins |
| IDTs | Industrial Digital Twins |
| IoT | Internet of Things |
| RAMI | Reference Architectural Model Industry 4.0 |
| I4.0 | Industry4.0 |
| SME | Small and Medium enterprise |
| PLC | Programmable Logic Controller |
| TRL | Technology Readiness Level |
| DBMS | Data Base Management System |

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
