# Peer review of "IoTwins: Implementing Distributed and Hybrid Digital Twins in Industrial Manufacturing and Facility Management Settings"

_futureinternet, doi:10.3390/fi16020065_

Round 1

Reviewer 1 Report

Comments and Suggestions for Authors

This paper presents some of the results of the IoTwins EU Horizon 2020 project, which dealt with the very popular issue in recent years of Digital Twins in industrial environments. The paper is relevant to the journal and its readers. Overall, I found the paper interesting. Although not groundbreaking, it presents a reference architecture for DTs in such environments, along with some description and high-level architecture of the approach implemented within the project. 

Although as I mentioned the paper is not groundbreaking, since it follows existing concepts and architectures and utilizes existing popular tools, it is very interesting to see how the high-level concepts and reference architectures were adopted in this project, which dealt with a number of different use-cases, as evidenced by the number of different testbeds mentioned in the paper.

What would have been very helpful for the readers, and the community in general, would be to include some "lessons learned" part in the text, to understand what were the conditions that led to adopting the design and implementation decisions in the project, since the main contribution of the paper is in the reference architecture and the implementation examples described. Maybe Section 6 would be the part of the paper where such a text would better fit. I would also suggest to add some additional text indicating the differentiating factors of industrial DTs compared with other sectors, i.e., why this architecture and implementation is a good fit and not another one. These aspects are not developed in the text currently.

In line 62, the phrase "The idea of the DT was formulated by Michael Grieves and introduced back in 2003" I think is a bit too strong. I think the authors mean the first use of the term "digital twin", because the concept was more or less used several years before the paper of Grieves (e.g., by NASA). 

Regarding definitions and concepts about the use of digital twins in industry, I would suggest that the authors also take a look at the following paper, which deals with several of the concepts discussed here and their definitions:

C. Koulamas and A. Kalogeras, "Cyber-Physical Systems and Digital Twins in the Industrial Internet of Things [Cyber-Physical Systems]," in Computer, vol. 51, no. 11, pp. 95-98, Nov. 2018, doi: 10.1109/MC.2018.2876181.

Reference [5] maybe could be changed (to be more accurate) to:

P. Adolfs, et al., “Reference Architecture Model Industrie 4.0 (RAMI4.0),” Deutsche Institut für Normung e.V. (DIN), Berlin, Germany, DIN SPEC 91345, 2016

Also maybe the terms "Industrie 4.0" (wrt. RAMI) and "Society 5.0" could be mentioned in the introduction as well, since they are very relevant to this work, and discussed within this context.

Regarding the presentation of the paper, it is well-written and easy to follow. There are however a few minor grammar or syntax mistakes in the text that need to be checked, e.g.  in lines 52 and 141 "inspired to" should be "inspired by", or in line 55 "is as follows. in Section" something is off. In line 180, reference to Table 1 is broken. The figures of the paper are of good quality and enhance the presentation of this work, but maybe the font in some cases, like in Figure 1, could be a bit bigger in some parts.

Overall, the paper is quite interesting and I recommend to accept it with a few minor changes, as described in the comments above.

Author Response

Let me take this opportunity to thank you for your useful comments. As requested by the Editor, a point-by-point response has been provided. To simplify the referees’ work, main modifications have been highlighted in yellow in the manuscript.

This paper presents some of the results of the IoTwins EU Horizon 2020 project, which dealt with the very popular issue in recent years of Digital Twins in industrial environments. The paper is relevant to the journal and its readers. Overall, I found the paper interesting. Although not groundbreaking, it presents a reference architecture for DTs in such environments, along with some description and high-level architecture of the approach implemented within the project.

Although as I mentioned the paper is not groundbreaking, since it follows existing concepts and architectures and utilizes existing popular tools, it is very interesting to see how the high-level concepts and reference architectures were adopted in this project, which dealt with a number of different use-cases, as evidenced by the number of different testbeds mentioned in the paper.

Comment 1.1

What would have been very helpful for the readers, and the community in general, would be to include some "lessons learned" part in the text, to understand what were the conditions that led to adopting the design and implementation decisions in the project, since the main contribution of the paper is in the reference architecture and the implementation examples described. Maybe Section 6 would be the part of the paper where such a text would better fit. I would also suggest to add some additional text indicating the differentiating factors of industrial DTs compared with other sectors, i.e., why this architecture and implementation is a good fit and not another one. These aspects are not developed in the text currently.

Response 1.1

Thank you for the valuable comment. Reading once again the paper, we realized that such an information gap exists. We added new text in Sections 4 and 5 to better motivate the choices concerning the architectural design and the software implementation.

Comment 1.2

In line 62, the phrase "The idea of the DT was formulated by Michael Grieves and introduced back in 2003" I think is a bit too strong. I think the authors mean the first use of the term "digital twin", because the concept was more or less used several years before the paper of Grieves (e.g., by NASA). 

Response 1.2

We rephrased the sentence following your advice.

Comment 1.3

Regarding definitions and concepts about the use of digital twins in industry, I would suggest that the authors also take a look at the following paper, which deals with several of the concepts discussed here and their definitions:

Koulamas and A. Kalogeras, "Cyber-Physical Systems and Digital Twins in the Industrial Internet of Things [Cyber-Physical Systems]," in Computer, vol. 51, no. 11, pp. 95-98, Nov. 2018, doi: 10.1109/MC.2018.2876181.

Reference [5] maybe could be changed (to be more accurate) to:

Adolfs, et al., “Reference Architecture Model Industrie 4.0 (RAMI4.0),” Deutsche Institut für Normung e.V. (DIN), Berlin, Germany, DIN SPEC 91345, 2016

Response 1.3

Thank you for the pointer. We found it interesting, so we decided to reference it in Section 2. Reference [5] was adjusted as per your suggestion.

Comment 1.4

Also maybe the terms "Industrie 4.0" (wrt. RAMI) and "Society 5.0" could be mentioned in the introduction as well, since they are very relevant to this work, and discussed within this context.

Response 1.4

Industrie 4.0 is now mentioned in the Abstract. The introductory section already mentioned it.

Comment 1.5

Regarding the presentation of the paper, it is well-written and easy to follow. There are however a few minor grammar or syntax mistakes in the text that need to be checked, e.g.  in lines 52 and 141 "inspired to" should be "inspired by", or in line 55 "is as follows. in Section" something is off. In line 180, reference to Table 1 is broken. The figures of the paper are of good quality and enhance the presentation of this work, but maybe the font in some cases, like in Figure 1, could be a bit bigger in some parts.

Response 1.5

Thank you for noticing the mistakes. They have all been fixed. We also improved the quality of figure 1.

Overall, the paper is quite interesting and I recommend to accept it with a few minor changes, as described in the comments above.

Reviewer 2 Report

Comments and Suggestions for Authors

The submitted paper IoTwins: Implementing Distributed and Hybrid Digital Twins inIndustrial Manufacturing and Facility Management Settings presents interesting, innovative and relevant perspectives on the design and testing of industrial digital twins.

Introduction - clearly presented with emphasis on the problems addressed and current competing solutions.

Background and Related Work - using a sufficient number of relevant sources, the current state of knowledge in the field is described with a loose relationship to the following chapter Design of a Hybrid and Distributed Digital Twin Model for Industrial Scenarios.

This includes the design of a hybrid and distributed digital twin model for industrial systems and their corresponding scenarios. The designs are relevant, well described and justified. 

Note the reference in line 180, which is incorrect.

A RAMI-inspired reference architecture - describes the Industry 4.0 (RAMI 4.0) reference architecture model used as inspiration for the design of the new solution. The following chapter describes the software prototype, followed by a description of the creation and deployment of the Industrial Digital Twin. Both parts are described in a clear and concise manner, with an emphasis on the innovative nature of the solution.

The conclusion is then too short and does not refer to current and future applications. This needs to be elaborated. Summarise the main benefits of the proposal and the results of the tests. 

Author Response

Let me take this opportunity to thank you for your useful comments. As requested by the Editor, a point-by-point response has been provided. To simplify the referees’ work, main modifications have been highlighted in yellow in the manuscript.

The submitted paper IoTwins: Implementing Distributed and Hybrid Digital Twins inIndustrial Manufacturing and Facility Management Settings presents interesting, innovative and relevant perspectives on the design and testing of industrial digital twins.

Introduction - clearly presented with emphasis on the problems addressed and current competing solutions.

Background and Related Work - using a sufficient number of relevant sources, the current state of knowledge in the field is described with a loose relationship to the following chapter Design of a Hybrid and Distributed Digital Twin Model for Industrial Scenarios.

This includes the design of a hybrid and distributed digital twin model for industrial systems and their corresponding scenarios. The designs are relevant, well described and justified. 

Comment 2.1

Note the reference in line 180, which is incorrect.

Response 2.1

Noted and fixed. Thank you.

A RAMI-inspired reference architecture - describes the Industry 4.0 (RAMI 4.0) reference architecture model used as inspiration for the design of the new solution. The following chapter describes the software prototype, followed by a description of the creation and deployment of the Industrial Digital Twin. Both parts are described in a clear and concise manner, with an emphasis on the innovative nature of the solution.

Comment 2.2

The conclusion is then too short and does not refer to current and future applications. This needs to be elaborated. Summarise the main benefits of the proposal and the results of the tests.

Response 2.2

Thank you for the comment. We expanded the conclusion, which now accounts for your valuable suggestion.